# Effects of a Cellulose Aerogel Template on the Preparation and Adsorption Properties of Coal Gangue-Based Multistage Porous ZSM−5

**DOI:** 10.3390/ma16113896

**Published:** 2023-05-23

**Authors:** Xue Ma, Chengli Ding, Hongsheng Yang, Xiao Zhu

**Affiliations:** Key Laboratory of Coal Clean Conversion & Chemical Engineering Process, School of Chemical Engineering, Xinjiang University, Urumqi 830046, China; maxue@stu.xju.edu.cn (X.M.); yanghongsheng@stu.xju.edu.cn (H.Y.); zhuxiao@stu.xju.edu.cn (X.Z.)

**Keywords:** coal gangue, cellulose aerogel, zeolite molecular sieves, malachite green

## Abstract

In this study, a ZSM−5/CLCA molecular sieve was prepared by the hydrothermal method using coal gangue as the raw material and cellulose aerogel (CLCA) as the green templating agent, which not only reduces the cost of traditional molecular preparation but also improves the comprehensive resource utilization rate of coal gangue. Through a series of characterization methods (XRD, SEM, FT-IR, TEM, TG, and BET), the crystal form, morphology, and specific surface area of the prepared sample were tested and analyzed. The performance of the adsorption process of malachite green (MG) solution was analyzed by adsorption kinetics and adsorption isotherm. The results show that the synthesized zeolite molecular sieve and the commercial zeolite molecular sieve are highly consistent. At a crystallization time of 16 h, a crystallization temperature of 180 °C, and an additive amount of cellulose aerogel of 0.6 g, the adsorption capacity of ZSM−5/CLCA for MG was up to 136.5 mg/g, much higher than that of commercially available ZSM−5. This provides an idea for the green preparation of gangue-based zeolite molecular sieves to remove organic pollutants from water. Moreover, the process of adsorbing MG on the multistage porous molecular sieve, which is spontaneous, conforms to the pseudo-second-order kinetic equation and Langmuir isothermal adsorption model.

## 1. Introduction

Water pollution is one of the world’s major challenges, especially when it is caused by organic dyes in printing and dyeing wastewater [1]. The continued accumulation of such pollutants can seriously endanger the safety of water resources. At present, the main methods commonly used to treat dyeing wastewater in China are adsorption, membrane separation, and chemical precipitation. Among them, the adsorption method is considered one of the most effective due to its simple operation and high efficiency in treatment [2]. However, traditional adsorbents have encountered obstacles in practical applications due to high preparation costs and difficulties in recovery. Therefore, finding a low-cost adsorbent material with excellent adsorption performance that is friendly to the environment is the primary challenge.

Gnanamoorthy et al. [3] amine-functionalized Bi_2_Sn_2_O_7_/rGO nanocomposites were prepared for the degradation of methylene blue by using thermal decomposition and in situ methods. The experimental results demonstrated that the photocatalytic performance of AF-Bi_2_Sn_2_O_7_/rGO nanocomposites was superior to that of Bi_2_Sn_2_O_7_ and AF-Bi_2_Sn_2_O_7_. Gnanamoorthy et al. [4] prepared CuNiO_2_ and CuNiO_2_/rGO nanocomposites by a hydrothermal method, which showed good photocatalytic properties for methylene blue. Bilgic et al. [5] successfully immobilized BODIP derivatives on APTES-modified sporopollenin microcapsules to obtain a fluorescent microcapsule sensor (Sp-APTES-monoBODIPY) for the selectivity and removal of Cu (II). The experimental results showed that Sp-APTES-monoBODIPY could be used as an ideal adsorbent and sensor material for Cu (II) with a maximum adsorption capacity of 25 mg/g by the complex quasi-secondary kinetic model and Langmuir model. Qu et al. [6] chose four different commercially available activated carbons to test the adsorption performance of MG. The experimental results showed that the four activated carbons conformed to a quasi-second order kinetic model at the beginning of adsorption; with increasing adsorption, the adsorption followed a quasi-first order equation, and the adsorption mechanism belonged to single-molecule physical adsorption.

Xinjiang is one of China’s largest coal fields, with an annual coal yield of nearly 400 million tons. Coal gangue (CG) is produced during coal preparation and mining, accounting for about 10–15% of the total coal production. The accumulation of CG reduces the utilization rate of available land resources, induces geological disasters, and pollutes the air and water, seriously threatening human health [7]. For the moment, the overall comprehensive utilization efficiency of CG is relatively low. CG is partially used in building materials [8], agricultural production [9], fill reclamation, power generation, and other industries [10]. Therefore, it is very meaningful to study CG’s comprehensive higher-value utilization as the raw material for constructing a clean, low-carbon, safe, and efficient coal industry system and forming a coal mine development pattern of harmonious coexistence between humans and nature.

CG can be used as the material for molecular sieves because of the high content of Al_2_O_3_ and SiO_2_ [11]. As the core catalytic material in several crucial processes represented by petroleum refining, molecular sieve materials are essential to resource transformation and environmental protection, playing an irreplaceable role in the sustainable development of the national economy. However, the industrial production of traditional molecular sieve materials is a typical route of high pollution and energy consumption, with a large discharge of three wastes. It is one of the industry’s urgently needed green upgrades in China’s current environmental protection strategy [12]. After proper treatment, CG can be used as the raw industrial material for cheap molecular sieve adsorbents. Exploring new uses of CG has become a research hotspot.

With corn starch as the pore-forming agent and CG and bauxite as the raw materials, Lu et al. [13] successfully prepared porous mullite ceramic carriers for filter membranes under a sintering temperature of 1100–1500 °C. They systematically studied the dynamic sintering behavior, phase evolution, shrinkage, porosity and pore radius, gas permeation flux, microstructure, and mechanical properties of the material, providing a new way to utilize CG comprehensively. Jablonska and Siedlecka [14] carried out adsorption tests on organic substances and heavy metals in industrial wastewater with adsorption materials prepared from CG. The results showed that the adsorption capacity of phenol (phenol concentration up to 10 mg/dm^3^) was about 50%, and the adsorption rate of heavy metals Pb (II), Ni (II), and Cu (II) was more than 70%. Jin et al. [15] prepared a new type of CG-based composite adsorption material with CG as the raw material by the method of alkali activation hydrothermal. By investigating the optimal conditions and reaction mechanisms of product synthesis, they proved that the material had good adsorption performance for Pb^2+^. Zhang et al. [16] studied synthesizing calcium silicate hydrates from CG to remove aqueous solution and Cr (VI) and Cu (II) in aqueous solution. The optimum synthesis parameters were 700 °C, 1 h, and a Ca/Si molar ratio of 1.0. The prepared samples were subjected to quantitative adsorption tests and adsorption mechanism analysis under different conditions, such as temperature, dosage, solution pH, initial metal concentration, and reaction time. The research showed that with CG as the raw material, an environmentally friendly adsorbent calcium silicate hydrate could be prepared, effectively removing metals from aqueous solutions through different mechanisms. Kong et al. [17] studied the extraction of iron and aluminum from CG. Liang et al. [18] synthesized NaX zeolite from CG and explored its removal effect and mechanism of Cd^2+^ and Cu^2+^. However, the synthesis process and conditions were not given. Sun et al. [19] prepared a ceramsite adsorbent that could adsorb copper ions continuously and efficiently with CG, pulverized coal, and copper slag as the main raw materials, and the maximum adsorption capacity could reach 20.6 mg/g. Bu et al. [20] and Ge et al. [21] studied the methods of synthesizing NaY molecular sieves by alkali fusion and hydrothermal with CG as the raw material, respectively. The synthesized NaY had good adsorption performance for lead (Pb^2+^), and the removal rate was as high as 100%. After five adsorption/desorption cycles, the removal rate was still over 63.71%. Zhao et al. [22] prepared a CG-rape straw biochar (CG-RS) composite from CG and rape straw to treat chromium-containing wastewater. The experiment showed that the composite material could effectively remove chromium from wastewater. To sum up, a few scholars have reported that cellulose aerogel was used as a template for the synthesis of zeolite molecular sieves to treat CG in previous studies.

According to commercial zeolite molecular sieves, this paper provides a new method of using cellulose green template and solvent in the synthesis of solid waste CG-based molecular sieves. To realize the resource utilization of wastes, CG is used as the raw material to prepare molecular sieve environmental remediation materials, which greatly reduces the discharge of three wastes and improves the comprehensive utilization of solid wastes. By adjusting the ratio of silicon to aluminum in CG, the CG-based multistage porous molecular sieve material with high crystallinity was successfully synthesized by the method of hydrothermal crystallization-calcination. This study discussed the potential of CG with a high content of Al and Si as the raw material to prepare zeolite molecular sieves, and the optimum technological parameters for the synthesis of zeolite molecular sieves were determined. Through a series of adsorption experiments with different parameters (such as initial solution, pH, etc.), the interaction of the CG-based zeolite molecular sieve in the adsorption of malachite green (MG) in wastewater was determined, and the adsorption kinetics and isothermal model were studied. Finally, the application prospect of CG adsorption in dye-polluted environments was expounded.

## 2. Experiment Materials and Methods

### 2.1. Experiment Materials

After crushing the CG (Yankuang Xinjiang Mining Co., Ltd., Zoucheng, China, sulfur ditch coal mine), it was analyzed and SiO_2_ and Al_2_O_3_ were found to be the main chemical components, accounting for 64.49%. They are the basic materials for the preparation of various zeolite molecular sieves, indicating the great potential of CG as the material for synthesizing zeolite molecular sieves. Additionally, CG also contains a certain amount of other metal oxides, such as Fe_2_O_3_, CaO, and MgO, which may affect the surface chemical properties and adsorption properties of the material.

Cotton linter cellulose (Xinjiang Aksu cotton linter processing plant, Xinjiang, China); Sodium hydroxide (NaOH, Tianjin Zhiyuan Chemical Reagent Co., Ltd., Tianjin, China); Tetraethyl orthosilicate (TEOS, Tianjin Zhiyuan Chemical Reagent Co., Ltd., Tianjin, China); Sodium aluminate, ammonia, methylamine (NaAlO_2_, NH_3_·H_2_O, CH_3_NH_2_, Shanghai McLean Biochemical Technology Co., Ltd., Shanghai, China); and Malachite Green (MG, Tianjin Tianxin Fine Chemical Development Center, Tianjin, China) are all commercially available materials and can be used directly in the sample preparation process.

### 2.2. Synthesis of ZSM−5/CLCA

Cellulose aerogels (CLCA) not only have excellent properties such as low density, high porosity, and specific surface area, but the hydroxyl functional groups on their surfaces can interact with silica-aluminum gels and act as structural guides, allowing for changes in the specific surface area and pore volume of the sample. For the reparation of CLCA, 4 g of cellulose powder was dispersed in NaOH/Urea/H_2_O (7:12:81 by mass) and then placed in a refrigerator for 12 h. The samples were then allowed to thaw at room temperature, washed to neutrality with deionized water, and freeze-dried for 24 h to prepare cellulose aerogel (CLCA).

Reference [23]. The steps for the synthesis of ZSM−5/CLCA molecular sieves are shown in Figure 1. Appropriate amounts of alkali-pretreated CG, deionized water, ammonia, methylamine, and NaOH were weighed, placed into a beaker, and stirred vigorously until the solution was fully mixed. Then, TEOS was added to the mixed solution (the aim was to regulate the molecular silicon-to-aluminum ratio) and stirred at room temperature to form a lotion. Then, the prepared CLCA was added to the above solution and stirred at room temperature to disperse evenly. Then, the lotion was poured into a 100-mL polytetrafluoroethylene reactor, and the reactor was put into an oven. After crystallization at a certain temperature for some time, the reactor was removed from the oven and left to cool naturally. The crystallization solution was removed from the tank and washed to neutralize it with deionized water. Then, it was dried overnight in an oven at a temperature of 60 °C. Finally, the product was ground, heated in a tubular furnace at a rate of 5 °C/min to 550 °C, and baked for 6 h. Then, the template was removed from the crystal, and the resulting sample was named ZSM−5/CLCA.

### 2.3. Characterization Method

An X-ray diffractometer (XRD, Bruker D8 Advance, Karlsruhe, Germany) was used to analyze the characteristics of the crystal form and the crystallization of the sample. The test conditions are: the sample was powdered (in line with the test requirements, so there was no need to grind before the test), Cu target Kα radiation (λ = 1.5406 Å), scanning angle range of 5~40°, scanning speed of 5°/min (the grinding mentioned in the preparation process was mortar grinding). The purpose was to fully disperse the particles; cellulose aerogel as a template only plays a role in pore-making, and then the templating agent in the crystal needs to be removed by roasting. Therefore, it will not affect the crystallinity of the sample). The appearance of the sample was observed by the scanning electron microscope (SEM, Germany ZSISS Sigma 300, Oberkochen, Germany) test process: A trace sample was directly glued to the conductive adhesive, and the Oxford Quorum SC7620 sputtering coater was used to spray gold for 45 s at 10 mA. Then, the sample morphology was photographed at a 3 kV acceleration voltage. A Fourier transform infrared spectrometer (FT-IR, Bruker VERTEX70, Karlsruhe, Germany) was used to conduct the characterization analysis of the functional groups of the sample. The specific surface area and void distribution of the sample were calculated for the characterization analysis using the automatic specific surface area and porosity analyzer (BET, America Micromerics ASAP 2460, Norcross, GA, USA). The thermal stability of the samples was analyzed using a thermogravimetric analyzer (TG, TA TGA 550, New Castle, DE, USA) under the following conditions: The samples were measured at a temperature increase rate of 10 °C/min from room temperature to 800 °C. The morphology of the molecular sieve was analyzed using a high-power transmission electron microscope (TEM, JEM-2100, Tokyo, Japan). The test was carried out by dispersing the sample powder in anhydrous ethanol, then dropping the suspension onto a copper grid and waiting for the ethanol to evaporate.

### 2.4. Adsorption Experiment

Next, we explored the adsorption performance of ZSM−5/CLCA (compared with commercial ZSM−5). The adsorption effects of the adsorbent on MG were investigated under different conditions, such as time, initial concentration of MG, temperature, and pH value. After adsorption, the 25 × 0.45 μm organic filter was used as a membrane to separate the adsorbent from the MG solution. The UV-visible spectrophotometer was used to record the absorbance of the solution before and after adsorption. The adsorption capacity and removal rate of the adsorbent from MG were calculated according to Formulas (1) and (2).

Adsorption capacity:(1)$$q_{t}=\frac{\left(C_{0}-C_{t}\right)V}{W}$$

Removal rate:(2)$$\eta =\frac{\left(C_{0}-C_{t}\right)}{C_{0}}\times 100\;\%$$
where q_t_ is the adsorption capacity, unit: mg/g; C_0_ represents the initial concentration of MG solution, and C_t_ represents the concentration of MG solution at time t, unit: mg/L; V is the volume of MG solution, unit: L; W is the mass of adsorbent, unit: g; η represents the removal rate, unit: %.

## 3. Results and Discussion

### 3.1. Effects of Preparation Conditions on the Synthesis of ZSM−5/CLCA

#### 3.1.1. Effects of Crystallization Time on the Synthesis of ZSM−5/CLCA

Figure 2a,b show that at 3 h, the sample does not have the characteristic diffraction peak of ZSM−5. Because the crystallization time is short, the molecular sieve is still in the initial induction period and has not formed the zeolite molecular sieve framework. When the crystallization time is extended to 5 h, the sample shows the unique five-finger peak shape of the ZSM−5 molecular sieve. At 2θ = 7.94°, 8.87°, 23.08°, 23.31° and 23.96°, all have the same characteristic diffraction peaks, which correspond to the characteristic peaks of (101), (200), (332), (051), and (303) of the crystal plane of ZSM−5/CLCA, indicating that the sample enters the growth stage after 5 h of crystallization. When the crystallization time is > 16 h, the intensity of the characteristic diffraction peak of ZSM−5/CLCA does not change significantly, indicating that the crystal skeleton growth has basically ended at this time.

The relative crystalline pairs are calculated as a percentage of the sample’s (2θ) = 7.5–24° diffraction peak area to the reference sample’s 7.5–24° diffraction peak area. In this paper, a sample with good crystallinity (crystallization time of 20 h, crystallization temperature of 180 °C, and CLCA of 0.6 g) was chosen as the reference sample for the calculation of crystallization pairs, which were calculated as:$$\mathrm{Relative}\;\mathrm{crystallinity}\;(\%)=\;\frac{\mathrm{Peak}\;\mathrm{area}\;\mathrm{between}\;2\theta =7.5–24{}^{\circ}\;\mathrm{of}\;\mathrm{the}\;\mathrm{solid}\;\mathrm{product}}{\mathrm{Peak}\;\mathrm{area}\;\mathrm{between}\;2\theta =7.5–24{}^{\circ}\;\mathrm{of}\;\mathrm{the}\;\mathrm{reference}\;\mathrm{sample}}$$

As can be seen from Figure 2c, when the crystallization time is 16 h, the crystallinity of the sample is up to 89%. If the crystallization time is prolonged, the crystallinity of the sample decreases gradually [24]. Therefore, the crystallization time of 16 h is the optimum time.

#### 3.1.2. Effects of Crystallization Temperatures on the Synthesis of ZSM−5/CLCA

XRD analyzed the ZSM−5/CLCA molecular sieve samples synthesized at different temperatures, and the XRD spectrum of Figure 3a was obtained.

Figure 3a,b show that when the crystallization temperature is 100 °C, the sample presents amorphous silica spectrum peaks, indicating that molecular sieves cannot be synthesized at this crystallization temperature. When the crystallization temperature rises to 120 °C, the sample has weak characteristic ZSM−5 diffraction peaks at 2θ = 7–23°, indicating that at this temperature, ZSM−5/CLCA is at the initial stage of the formation of the molecular sieve skeleton structure and crystal grains are gradually forming. However, the reaction is relatively slow because of the low temperature. With the increasing crystallization temperature, the peak shape is high and sharp, and the peak width gradually narrows. After the temperature reaches 180 °C, the intensity and crystallinity of the characteristic diffraction peaks of the prepared samples decrease with the increase in temperature.

Figure 3c shows that when the crystallization temperature is 120 °C, 140 °C, 160 °C, 170 °C, 180 °C, and 190 °C, the crystallinity is 10%, 39%, 65%, 70%, 92%, and 80%, respectively. This paper selects the highest crystallinity as the crystallization temperature of the synthetic sample; that is, the crystallization temperature is 180 °C.

Figure 4 shows the SEM diagram of the ZSM−5/CLCA molecular sieve synthesized at different crystallization temperatures. It can be seen that when the crystallization temperature is 100 °C, there are only amorphous crystals, consistent with the XRD characterization results in Figure 3a. When the crystallization temperature is 140 °C, the surface of the sample is crowded with small particles, which is caused by the incomplete crystallization of amorphous aluminosilicate. When the crystallization temperature is 180 °C, it can be clearly observed that the crystal surface is smooth and the size is relatively uniform, showing a typical “coffin” structure. If the crystallization temperature continues to increase, cracks and holes will appear on the surface of some samples, leading to the collapse of the skeleton.

#### 3.1.3. Effects of Cellulose Aerogel Addition on the Synthesis of ZSM−5/CLCA

The template plays an important role in the hydrothermal synthesis of molecular sieves, such as in structure orientation, skeleton space-filling, and the balance of skeleton charge. The amount of the template’s addition affects the physicochemical and adsorption properties of molecular sieves [25].

As shown in Figure 5a, different amounts of CLCA addition have obvious characteristic diffraction peaks at 2θ = 7.89°, 8.80°, 23.03°, 23.28°, and 23.91°, respectively, indicating that ZSM−5/CLCA has been successfully prepared. When the amount of CLCA addition is 0.8 g, the intensity of the characteristic diffraction peak of ZSM−5/CLCA decreases, which may be due to the aggregation of molecules in the channel caused by the excessive addition of CLCA. Figure 5b shows the adsorption of MG by different amounts of CLCA addition. It can be seen that when CLCA is 0.1 g, 0.2 g, 0.4 g, 0.6 g, and 0.8 g, the adsorption capacity of the prepared sample to MG is 37 mg/g, 46 mg/g, 81 mg/g, 103 mg/g and 27 mg/g, respectively. Therefore, the amount of CLCA addition in this paper is 0.6 g.

#### 3.1.4. FT-IR and TG Analysis of ZSM−5/CLCA

Figure 6a shows the FT-IR diagram of the ZSM−5/CLCA molecular sieve. At 1233, 1104, 802, 552, and 448 cm^−1^, the characteristic skeleton vibration peaks of ZSM−5/CLCA appear. 1233 cm^−1^ and 1104 cm^−1^ represent the tensile vibration peak of the five-membered ring and the asymmetric stretching vibration peak of the Al-O-Si bond. At 800 cm^−1^, there is a symmetric stretching vibration peak of the Al-O-Si bond. 552 cm^−1^ and 448 cm^−1^ correspond to the typical characteristic peak of the double five-membered ring and T-O bond bending vibration of the MFI structure. This agreed with the results of previous studies [26].

The thermogravimetric curves of the ZSM−5/CLCA molecular sieve (CLCA = 0.6 g) are shown in Figure 6b. As can be seen from the graphs, the thermogravimetric curves of the ZSM−5/CLCA molecular sieves show two distinct weight loss processes and a total weight loss of 32.28%. The weight loss in the first stage was mainly caused by the removal of physically adsorbed water from the surface of the molecular sieve [27]; the weight loss in the second stage was attributed to the removal of bound and coordinated water from the molecular sieve and the precipitation of volatile fractions from the decomposition of the cellulose structure, releasing a large number of volatile gases [28]; the weight did not change in the third stage, which was probably due to the collapse of the molecular sieve skeleton and the appearance of a new phase. Therefore, the molecular sieve treatment temperature should be controlled below 560 °C.

#### 3.1.5. SEM and TEM Analysis of ZSM−5/CLCA

The SEM and TEM of the ZSM−5/CLCA molecular sieve at a crystallization time of 16 h, a crystallization temperature of 180 °C, and a CLCA of 0.6 g are shown in Figure 7. As can be seen, the sample has a flattened hexagonal prism shape with a rectangular structure inside the crystal.

#### 3.1.6. BET Analysis of ZSM−5/CLCA

Figure 8 shows that the curve presents two obvious hysteresis loops. At 0.2 < P/P0 < 0.5, the adsorption capacity increases with the relative pressure, which conforms to the basic characteristics of the type I isotherm. This may be because the interaction between N_2_ and the molecular sieve is enhanced in micropores, leading to micropore filling at low relative pressure [29]. At the same time, the molecular sieve has a typical Type IV adsorption and desorption curve. In the relative pressure P/P0 range of 0.4 to 1.0, the adsorption of molecular sieve showed an increasing trend, resulting in the adsorption-desorption isotherm not overlapping and the desorption isotherm being above the adsorption isotherm, thus making the nitrogen adsorption-desorption curve appear a clear hysteresis loop, which indicates that the molecular sieve has a certain mesoporous structure, and the hysteresis loop may be due to the occurrence of the molecular sieve mesopore channel This indicates that the molecular sieve has a mesoporous structure, and the hysteresis loop may be caused by capillary aggregation in the pore channels of the molecular sieve [30]. According to the Barrett-Joyner-Halenda (BJH) theory, the aperture distribution of the ZSM−5/CLCA molecular sieve is concentrated at 1–4 nm. It indicates that the multistage porous molecular sieve prepared with CLCA as the template still has microporous and mesoporous structures.

The aperture distribution of ZSM−5 and ZSM−5/CLCA molecular sieves was calculated using the BJH method. The specific data results are shown in Table 1. It can be seen that the specific surface area and total pore volume of ZSM−5 are 385 m^2^⋅g^−1^ and 0.1959 cm^3^⋅g^−1^, respectively, while those of ZSM−5/CLCA molecular sieve are 398 m^2^⋅g^−1^ and 0.2029 cm^3^⋅g^−1^, respectively. Through comparison, it is found that adding CLCA can increase the specific surface area and total pore volume of molecular sieves.

### 3.2. Adsorption Performance Analysis

#### 3.2.1. Effect of pH Value on Adsorption

MG is a kind of crystalline powder with a metallic luster. The color of its aqueous solution is affected by its pH value. When the pH value of the solution is less than 2.0, it is yellow-green. When the pH value is greater than 10.0, the color of the aqueous solution changes from blue-green to colorless, and MG is converted into the methanol-based molecular structure under the alkaline condition [31]. Therefore, the pH range in this paper is 3–9.

The effect of pH on the adsorption performance of the adsorbent was investigated by varying the initial pH of the MG solution (100 mg/L) at room temperature with a fixed adsorbent dosage of 0.05 g. Figure 9 shows that when the pH value is between 3 and 5, the adsorption capacity and removal rate of the adsorbent for MG are low. The possible reasons for this phenomenon are: on the one hand, when the solution pH is low, the skeletal structure of the molecular sieve is eroded by solid acid. On the other hand, it may be that under acidic conditions, H^+^ will compete with cationic dyes for adsorption sites [32]. When the pH value is between 6 and 9, the adsorption capacity and removal rate gradually increase until they reach equilibrium. This is because H^+^ decreases with the increase in pH value, and the competition between MG and H^+^ weakens, resulting in an increase in the adsorption capacity and removal rate. With continuous adsorption, the active sites on the surface of the molecular sieve are gradually occupied, resulting in the adsorption capacity and removal rate reaching saturation and coming to equilibrium. At this time, the adsorption capacity of ZSM−5 is 132 mg/g, with an adsorption rate of approximately 79%; the adsorption capacity of ZSM−5/CLCA molecular sieve is 164 mg/g, with an adsorption rate of approximately 98%.

#### 3.2.2. Effect of Different Initial Concentrations on Adsorption

This section focuses on the effect of different initial concentrations (20 mg/L, 40 mg/L, 60 mg/L, 80 mg/L, 100 mg/L, and 120 mg/L) on the adsorption performance of the molecular sieve. As can be seen from Figure 10, the adsorption of both molecular sieves increased with the increase in the initial concentration of MG. When the solution concentration increased from 20 mg/L to 120 mg/L, the adsorption of MG by ZSM−5 and ZSM−5/CLCA increased by 62.58 mg/g and 84.66 mg/g, respectively.

#### 3.2.3. Adsorption Kinetic Model

The adsorption kinetic model is one of the important parameters in the adsorption process that can reflect the adsorption rate [33]. Figure 11a shows that with the extension of adsorption time, the adsorption rate gradually slows down, and the adsorption reaches equilibrium in about 660 min, at which time the adsorption amount of ZSM−5 is 74.74 mg/g and that of ZSM−5/CLCA is 136.5 mg/g. According to Figure 11b,c, and Table 2 of fitting parameters, the adsorption of MG by molecular sieves more closely conforms to the pseudo-second-order kinetic model (the correlation coefficient R^2^ is 0.99 and 0.96, respectively). Figure 11d and Table 3 shows that the adsorption process can be divided into three stages: (1) MG spreads to the surface of the molecular sieve, that is, through external diffusion. In this stage, because of the large adsorption driving force and the large number of active sites on the surface of the molecular sieve, the adsorption rate is fast. (2) MG spreads from the outer surface of the molecular sieve to the pore channels of the inner surface, namely internal diffusion. (3) The adsorption and desorption rates of MG on the surface of the molecular sieve are the same, that is, they reach adsorption equilibrium [34]. The adsorption fitting curves of molecular sieves for MG do not pass the original point, indicating that internal diffusion is not the only control step and that multiple control steps determine the adsorption.

#### 3.2.4. Isothermal Adsorption Model

The adsorption isotherm model can reflect the relationship between the concentration of adsorbate in solid and liquid phases at a certain temperature. This paper uses the Langmuir isotherm adsorption model and the Freundlich isotherm adsorption model to fit the adsorption effect on MG. As shown in Figure 12 and Table 4, the fitting correlation coefficient R^2^ (0.98) of the Langmuir isotherm model is greater than that of the Freundlich isotherm adsorption model (0.73 and 0.46), that is, the Langmuir isotherm model can better describe the adsorption of MG by molecular sieves, and the adsorption process of MG by molecular sieves is single-molecular layer adsorption. In addition, the correlation coefficient R^2^ of the Temkin isothermal adsorption model was also high, indicating the existence of electrostatic solids and ion-exchange chemisorption in the adsorption process. Furthermore, the E values calculated by the D-R isothermal adsorption model were all greater than 16 KJ/mol, also indicating the existence of an ion-exchange mechanism in the adsorption process [35].

A comparison of the adsorption capacity of ZSM−5/CLCA with other adsorbent materials for MG is shown in Table 5. From the table, it can be found that ZSM−5/CLCA molecular sieves have a similar or better adsorption performance on MG. Therefore, ZSM−5/CLCA molecular sieves have the prospect of being an environmentally friendly adsorbent.

#### 3.2.5. Adsorption Thermodynamic Model

The experimental data were linearly fitted to the data using lnK_d_ as the vertical coordinate and 1/T as the horizontal coordinate to obtain Figure 13. The values of ΔS° and ΔH° were calculated using the van’t Hoff formula. Then the value of ΔG° was calculated by combining the third principle of thermodynamics, and the degree, direction, and feasibility of the adsorption reaction could be determined based on the positive and negative values of ΔS°, ΔH°, and ΔG° [39].

Table 6 shows the calculated values of ΔG°, ΔS°, and ΔH°. It can be concluded from Table 6 that ΔG° values were all in the range of −20 kJ/mol < ΔG° < 0 kJ/mol, indicating that the adsorption of MG by the molecular sieve was spontaneous physisorption [5]; ΔS° > 0 indicates the increase in chaos at the solid–liquid interface during adsorption [6]; ΔH° > 0 indicates that the process is an endothermic reaction, and raising the temperature is conducive to adsorption.

## 4. Conclusions

In this paper, a ZSM−5/CLCA multistage porous molecular sieve was prepared using the hydrothermal reaction method with CG as the raw material and CLCA as the template. Through analyzing the synthesis conditions, the following conclusions could be drawn:(1)Crystallization time, crystallization temperature, and the additive amount of template significantly affect the structure and properties of the ZSM−5/CLCA molecular sieve. When the optimum preparation conditions are: the crystallization time is 16 h, the crystallization temperature is 180 °C, and the additive volume of CLCA is 0.6 g, the adsorption of MG could reach 136.5 mg/g, which was much higher than the adsorption amount of the ZSM−5 molecular sieve for MG.(2)The results of the adsorption experiments show that the adsorption of MG by ZSM−5/CLCA is a spontaneous heat absorption reaction, which is in accordance with the quasi-secondary kinetic equation and the Langmuir isothermal adsorption model. By calculating the Temkind isothermal adsorption model and the D-R isothermal adsorption model, it is clear that the adsorption process involves electrostatic interaction and chemisorption by ion exchange.

## Figures and Tables

**Figure 1 materials-16-03896-f001:**
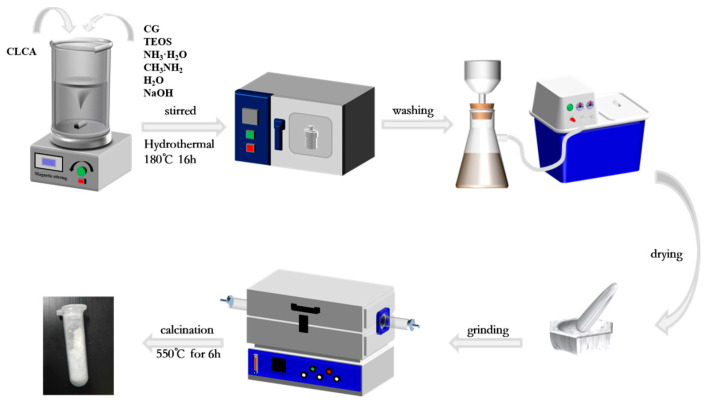
Preparation process of ZSM−5/CLCA molecular sieve.

**Figure 2 materials-16-03896-f002:**
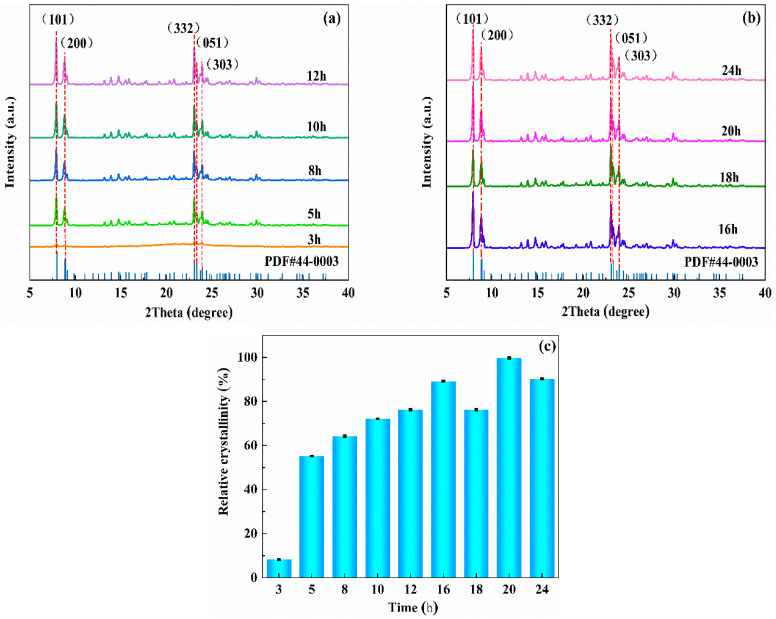
The diagrams of XRD (**a**,**b**) and crystallinity (**c**) of ZSM−5/CLCA synthesized at different crystallization times (crystallization temperature: 180 °C; CLCA: 0.6 g).

**Figure 3 materials-16-03896-f003:**
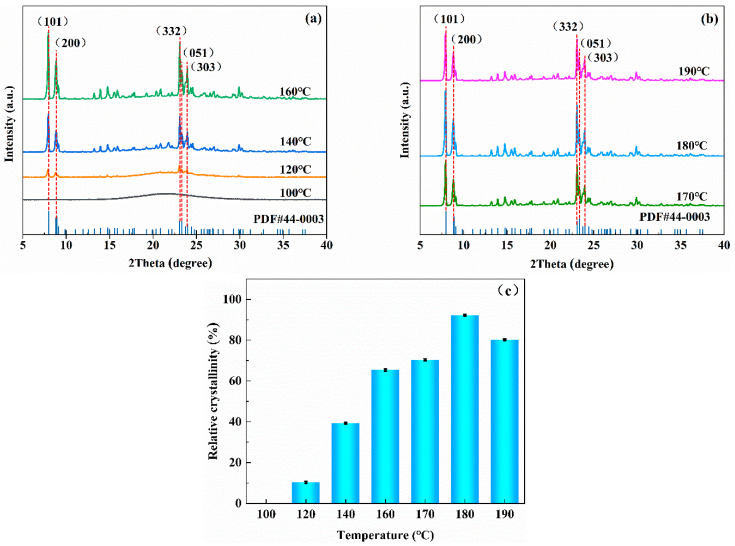
The diagrams of XRD (**a**,**b**) and crystallinity (**c**) of ZSM−5/CLCA molecular sieves synthesized at different crystallization temperatures (crystallization time: 16 h; CLCA: 0.6 g).

**Figure 4 materials-16-03896-f004:**
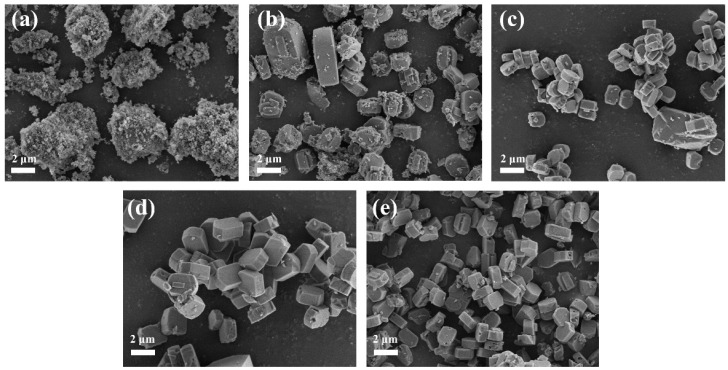
The SEM diagrams of ZSM−5/CLCA molecular sieves synthesized at different crystallization temperatures (crystallization time: 16 h; CLCA:0.6 g). (**a**) 100 °C (**b**) 140 °C (**c**) 170 °C (**d**) 180 °C (**e**) 190 °C.

**Figure 5 materials-16-03896-f005:**
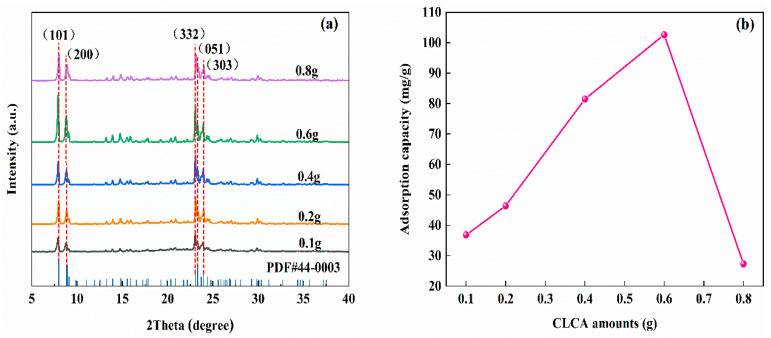
The XRD diagram (**a**) and MG adsorption diagram (**b**) of ZSM−5/CLCA molecular sieves synthesized with different amounts of CLCA addition (crystallization time: 16 h; crystallization temperature: 180 °C).

**Figure 6 materials-16-03896-f006:**
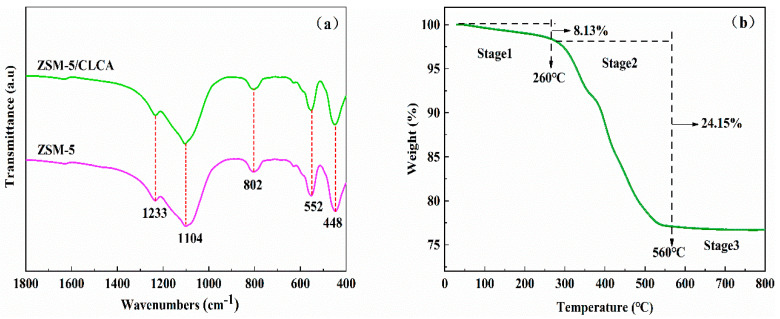
The Fourier transform infrared spectra of ZSM−5 and ZSM−5/CLCA (**a**), thermogravimetric analysis (**b**).

**Figure 7 materials-16-03896-f007:**
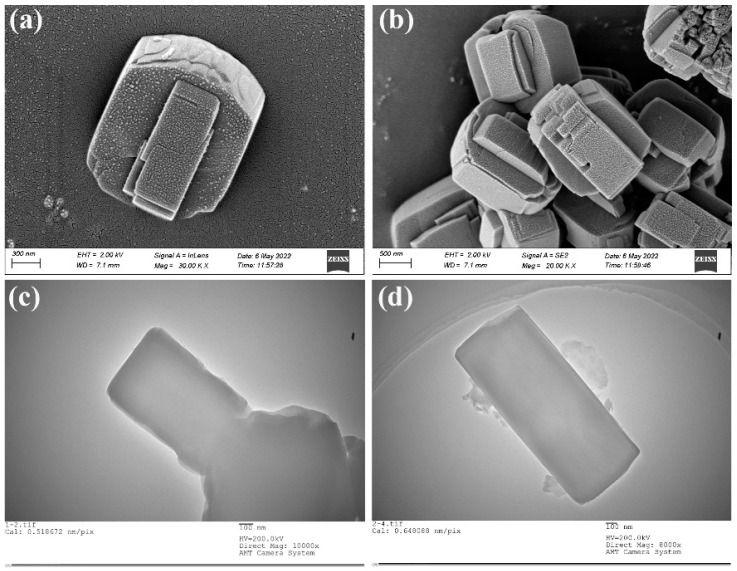
ZSM−5/CLCA molecular sieve SEM (**a**,**b**) and TEM (**c**,**d**).

**Figure 8 materials-16-03896-f008:**
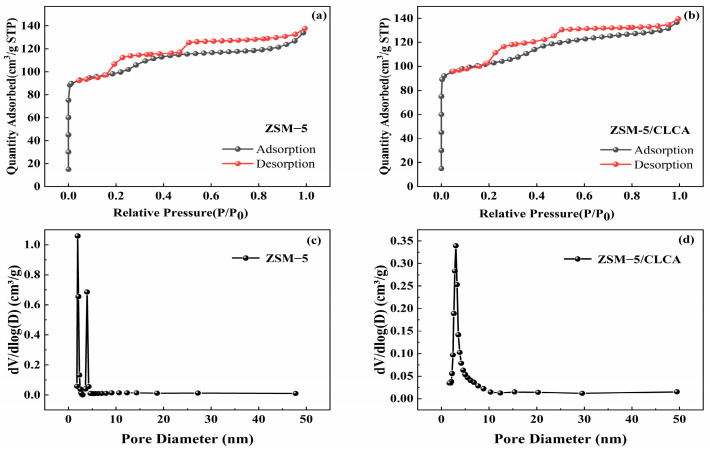
The N_2_ adsorption-desorption curve (**a**,**b**) and pore size distribution (**c**,**d**).

**Figure 9 materials-16-03896-f009:**
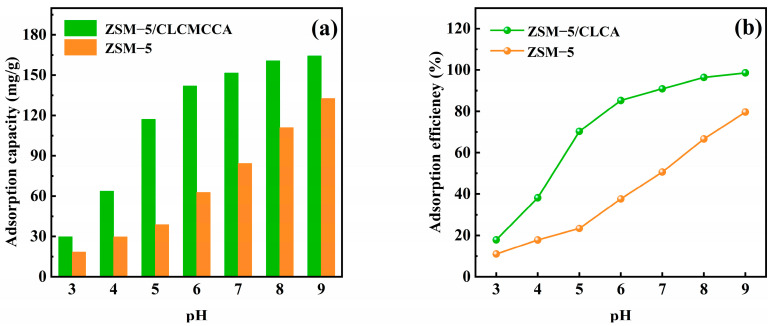
Effect of pH on the adsorption of MG by ZSM−5 and ZSM−5/CLCA. (**a**) adsorption amount, (**b**) adsorption rate.

**Figure 10 materials-16-03896-f010:**
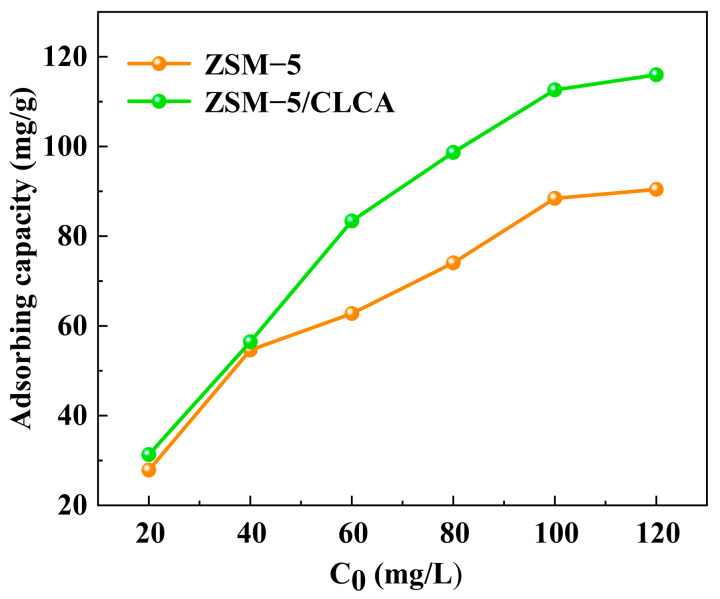
Effect of different initial concentrations on the adsorption performance of molecular sieves.

**Figure 11 materials-16-03896-f011:**
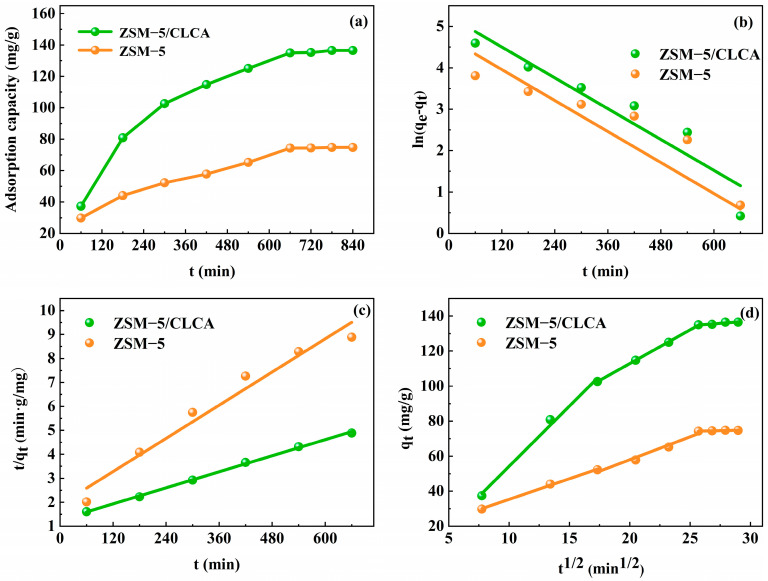
(**a**) the impact of time on MG; (**b**) pseudo-first-order dynamics fitting curve; (**c**) pseudo-second-order dynamics fitting curve; (**d**) the fitting curve of diffusion kinetics in particles.

**Figure 12 materials-16-03896-f012:**
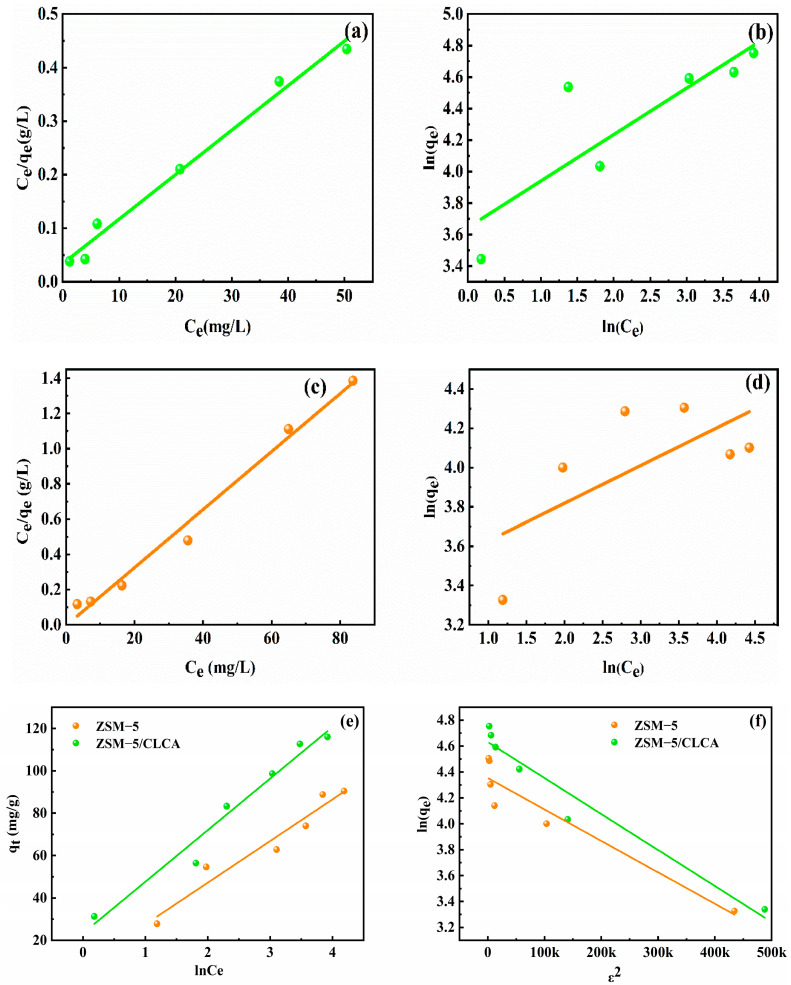
Isothermal adsorption fitted curves for two different molecular sieves adsorbing MG, (**a**,**c**) Langmuir isotherm, (**b**,**d**) Freundlich isotherm, (**e**) Temkin isotherm, (**f**) D-R isotherm.

**Figure 13 materials-16-03896-f013:**
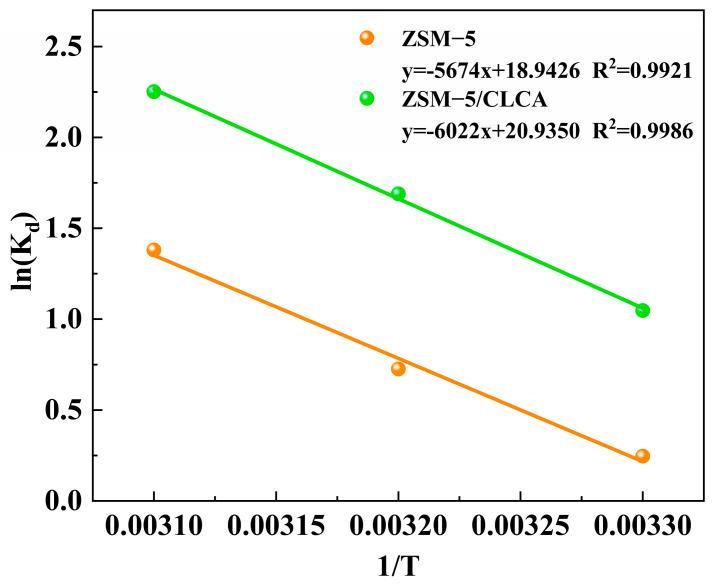
Van’t Hoff plots of two different molecular sieves for the adsorption of MG.

**Table 1 materials-16-03896-t001:** The BET characterization of ZSM−5 and ZSM−5/CLCA.

Sample	S_BET_ (m^2^⋅g^−1^)	S_mic_ (m^2^⋅g^−1^)	V_total_ (cm^3^⋅g^−1^)	V_mic_ (cm^3^⋅g^−1^)
ZSM−5	385	266	0.1959	0.1023
ZSM−5/CLCA	398	312	0.2029	0.1210

**Table 2 materials-16-03896-t002:** The kinetic adsorption parameters of ZSM−5 and ZSM−5/CLCA.

Sample	q_e,exp_ (mg/g)	Pseudo-First-Order Dynamic Parameters			Pseudo-Second-Order Dynamic Parameters		
		q_e1_ (mg/g)	k_1_	R^2^	q_e2_ (mg/g)	K_2_	R^2^
ZSM−5/CLCA	136.50	189.98	6.21 × 10^−3^	0.8955	178.57	2 × 10^−5^	0.9987
ZSM−5	74.78	110.73	6.24 × 10^−3^	0.7429	86.66	7 × 10^−5^	0.966

**Table 3 materials-16-03896-t003:** The Weber–Morris internal diffusion model parameters of ZSM−5 and ZSM−5/CLCA.

Sample	The First Stage		The Second Stage		The Third Stage	
	k_P1_	C_1_	k_P2_	C_2_	k_P3_	C_3_
ZSM−5/CLCA	6.8786	−14.6136	3.8504	35.8441	0.5302	121.2816
ZSM−5	2.3556	11.7786	2.6181	5.5893	0.1501	70.4392

**Table 4 materials-16-03896-t004:** Parameters for fitting isothermal adsorption models for two different molecular sieves adsorbing MG.

Sample	Langmuir Isothermal Adsorption Parameters			Freundlich Isothermal Adsorption Parameters		
	q_m_(mg/g)	K_L_	R^2^	1/n	K_F_	R^2^
ZSM−5/CLCA	120.37	0.2450	0.9863	0.2949	38.2889	0.7392
ZSM−5	60.75	0.3791	0.9871	0.1918	31.0441	0.4650
**Sample**	**Temkin Isotherm**			**D-R Isotherm**		
	**K_T_**	**b**	**R^2^**	**q_m_ (mg/g)**	**E (KJ/mol)**	**R^2^**
ZSM−5/CLCA	0.9588	101.87	0.9680	102.65	37.2726	0.9510
ZSM−5	0.4022	126.11	0.9441	77.75	34.8332	0.9064

**Table 5 materials-16-03896-t005:** Comparison of MB adsorption capacities with different adsorbent.

Adsorbent	q_e_ (mg/g)	Reference
Alg-Fe_3_O_4_	47.84	[36]
Cu-MOFs/Fe_3_O_4_	113.67	[37]
HA-Fe_3_O_4_	79.30	[38]
ZSM−5/CLCA	120.37	This work

**Table 6 materials-16-03896-t006:** The thermodynamic parameters of ZSM−5 and ZSM−5/CLCA at different temperatures.

Sample	T (K)	lnK_d_	ΔG° (kJ/mol)	ΔH° (kJ/mol)	ΔS° (J/mol/k)
ZSM−5/CLCA	303	1.0475	−2.6671	50.0711	174.0536
	313	1.6894	−4.4077		
	323	2.2520	−6.1482		
ZSM−5	303	0.2460	−0.5455	47.1736	157.4888
	313	0.7258	−2.1204		
	323	1.3809	−3.6953		

## Data Availability

Date will be made available on request.

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
