# Peer review of "Effects of a Cellulose Aerogel Template on the Preparation and Adsorption Properties of Coal Gangue-Based Multistage Porous ZSM−5"

_materials, 2023, doi:10.3390/ma16113896_

Round 1
Reviewer 1 Report
I think that this research paper is highly significant for the scientific society readers. It deals with a very hotspot issue that concern how to manage waste materials after coal mining and processing to perform molecular sieving materials, which are important for industry.
Reviewer 2 Report
Comments:
1. All figures resolution should be increased.
2. In a FTIR results, the author compared with previous reported results
3. In a figure captions, please mentioned a & b notations
4. We are recommending to compared other reported materials
5. As proposed materials with MG dye absorption studies are compare to other reported materials, as G. Gnanamoorthy, V. Karthikeyan, Daoud Ali, Gokhlesh Kumar, Virendra Kumar Yadav, V.Narayanan, Global popularization of CuNiO2 and their rGO nanocomposite loveabled to the photocatalytic properties of methylene blue, Environmental Research (2022) 112338. ;Photocatalytic properties of amine functionalized Bi2Sn2O7/rGO nanocomposites, G. Gnanamoorthy, S. Muthamizh, K. Sureshbabu, S. Munusamy, A. Padmanaban, A. Kaaviya, R. Nagarajan, A. Stephen, V. Narayanan, Journal of Physics and Chemistry of Solids,118 (2018) 21-31 ;
6. Please check grammatical mistakes and topographical errors.
7. This characteristic studies are not enough so please try to other studies like FESEM, TEM, Thermal analysis and etc.,

Reviewer 3 Report
Journal Name: Materials
Recommendation: Major
Manuscript Id: materials-2364579
Comments to Authors
The manuscript entitled, “Effects of Cellulose Aerogel Template on the Preparation and Adsorption Properties of Coal Gangue Based Multistage Porous ZSM-5” reported coal gangue based multistage porous zeolite molecular sieve and analyzed adsorption performance of the adsorption process of malachite green (MG) solution. This work is good but not well written and lacks some key points. Therefore, the publication of this work in its present form is not recommended for the publication in materials journal and can be justified after the authors consider the following points.
The following are some specific comments regarding the work covered in this manuscript.
1. In section 2.1, what do you mean by the statement “…..and do not need further treatment”. How you used these chemicals mentioned it clearly.
2. In section 2.2, the statement “Reference [17]” what do you mean by this? There is no clarity on how the material was synthesized. There are no specific reaction conditions mentioned. What do you mean by the statement “Added TEOS to the mixed solution for adjustment”? What kind of adjustment is there? When did you prepare the CLCA?
3. Details are missing in Figure captions and not organized well.
4. To optimize the temperature parameter for ZSM-5/CLCA at what amount of CLCA used in fig 3.1
5. Why FTIR spectrum reported only 400 to 1800 cm-1? Is there are no peaks beyond that limit or what? The functional group region starts after 1500 cm-1. What specific study the authors wants to explore using this limit? Any specific objective?
6. Same with adsorption experiments, the author has not mentioned any parameters like the amount of adsorbate and dye concentration at what the different effect was observing?
Given the above-mentioned points, the current version of the manuscript is recommended Major revision before the publication in the materials Journal.

Minor editing of English language required.
Reviewer 4 Report
1- Information should be given about the adsorption mechanism.
2- The adsorption mechanism should be mentioned more in the introduction. (https://doi.org/10.1016/j.cherd.2021.12.014)
3- If the Temkin and D-R isotherms of the adsorption isotherms are calculated, the chemical or physical properties of the adsorption can be better understood.
4- English should be improved.
5- Adsorption Thermodynamic Models should be calculated according to the following studies.
https://doi.org/10.1016/j.molliq.2020.113315
https://doi.org/10.1016/j.molliq.2018.10.048
Minor editing of English language required
Reviewer 5 Report
Utilizing coal gangue as starting materials for the synthesis of zeolite materials have already been reported but the novelty of this work lies on combining the use of the cellulose aerogel template in this kind of synthesis from waste material. Although the idea of the paper is alright, the way the manuscript was written is hard to read. In addition, a lot of data was not fully explained and the manuscript needs more additional discussion to justify the conclusion.
Comments:
1. The Al and Si content of coal gangue is not consistent. How would it affect the synthesis of the material?
2. Kindly indicate the temperature used for synthesis in section 3.1.1 when investigating the effect of crystallization time, as well as the reaction time used in section 3.1.2 when investigating the effect of crystallization temperature.
3. What caused the crystallinity of the synthesized material to drop at 18h, then increase again after 20h? How many times was this procedure done? Is it repeatable? Kindly show the error bars showing the standard deviations for the relative crystallinity of the synthesized materials using different reaction time (Figure 3-1c) to reveal whether 16h is really the optimal reaction time for synthesis.
4. In connection with the above comment, also show the error bars Figure 3-2c.
5. How was the relative crystallinity determined? Kindly show the equation and the basis of the computation.
6. It was mentioned: “If the crystallization temperature continues to be increased, cracks and holes will appear on the surface of some samples, leading to the collapse of the skeleton.” However, these “cracks and holes” are also observed in all the other SEM micrographs provided. Is it possible to quantify these cracks and holes and then correlate it with the decreased relative crystallinity? Kindly provide more evidence on how high temperature can cause this.
7. Mentioned in section 3.1.3: “It can be seen that when CLCA is 0.1g, 0.2g, 0.4g, 0.6g, and 0.8g, the adsorption capacity of the prepared sample to MG is 37mg/L, 46mg/L, 81mg/L, 27mg/L, and 103mg/L, respectively.” However, Figure 3-4b shows otherwise. Kindly confirm which is the correct data.
8. It was mentioned in section 3.1.5: “it is found that adding CLCA can increase the specific surface area and total pore volume of molecular sieves.” What properties of CLCA cause the increase in surface area and pore volume? Kindly explain.
9. How does the current material compare with other materials in published literature with regards to MG adsorption?
none
Reviewer 6 Report
Comments to the Author
This manuscript described a green approach to fabricate ZSM-5 zeolite and its application for the adsorption of a representative dye (malachite green) from water. The synthetic method and results for dye adsorption are meaningful and well represented in the MS. It can be acceptable for the publication after minor revisions following the comments below.
C1. An abstract must include the novelty of the research, that author general looks for, is lacking in the current abstract! Moreover, some quantitative results are preferable to understand the status of the results compare to the earlier reports.
C2. A significant problem/challenge in the relevant filed should be clearly explained in the introduction. It must be updated with good/suitable references. References should be enriched with high quality review/articles recently published in the relevant topic.
C3. Author may curios, how the crystallinity was estimated/measured? The XRD peak intensity may changes with a variety of factors like sample amount, sample thickness, scan etc. Therefore, method of crystallinity measurement should be explained clearly.
C4. For easy understanding of the synthetic method, an illustration/scheme is highly appreciated, if it is added in the revised MS.
C5. A representation (e.g. Table) to show the relative performance of the developed materials compare to the earlier reports can help reader to understand the status of the present study in the field.
C6. What is reason for higher adsorption capacity of ZSM-5/CLCA than commercial ZSM-5 even with comparable textural properties? That should be explained with proper supportive data.
C6. Typical errors should be minimized through careful check and revisions.
“The rate of removing MG exceeds 98%.”
-How rate can be 98%?
“The possible reasons for this phenomenon are: on the one hand, the low pH value leads to a strong acidity of the dye solution, and H+ may corrode some aluminosilicates in the zeolite and destroy the basic framework struc-ture of the molecular sieve.”
-Is seems something wrong in the sentences.
there may have many more like this……….
English should be checked/revised with an expart english editor!
Round 2
Reviewer 5 Report
The authors have carefully addressed the issues. It can be published now.
none